# Occupational burnout during war: The role of stress, disruptions in routine, sleep, work-family conflict, and organizational support as a moderator

**Maor Kalfon Hakhmigari**⬡<sup></sup>*, **Irene Diamant**⬡

School of Behavioral Sciences, The Academic College of Tel Aviv-Yafo, Tel Aviv, Israel

⬡ These authors contributed equally to this work.
* maorkalf@mta.ac.il

**Data Availability Statement:** All relevant data are within the manuscript

## Abstract

### Background

Occupational burnout, resulting from long-term exposure to work-related stressors, is a significant risk factor for both physical and mental health of employees. Most research on burnout focuses on routine situations, with less attention given to its causes and manifestations during prolonged national crises such as war. According to the Conservation of Resources theory, wartime conditions are associated with a loss of resources, leading to accelerated burnout. This study aimed to examine burnout among employees during the war in Israel that broke out in October 2023, placing the population nationwide in a state of existential threat and functional crisis. The researchers hypothesized that Stress, work-to-family conflict, family-to-work conflict, sleep disruption, and routine disruption (H1-H5) would be positively associated with occupational burnout. In addition, the study examined whether organizational strategies, such as flexibility and volunteering, would be negatively associated with burnout (H6) and whether they would moderate the relationship between work-family conflict and burnout during wartime (H7).

### Methods

The study involved 374 employees recruited through social networks via a snowball sampling method during the war. Participants completed a questionnaire assessing demographic details, burnout, stress, routine disruptions, work family conflict (work-to-family and family-to-work conflict), sleep disturbances, and organizational support measures. Hierarchical linear regression and Pearson correlation were utilized to model the outcome variable of occupational burnout and Process Model 1 was utilized for the moderation hypothesis.

### Results

Findings revealed a notable level of burnout during wartime (M = 3.95, SD = 1.15). Routine disruptions, work-to-family conflict, and sleep disturbances significantly contributed to burnout beyond stress levels, thus hypotheses H1 to H5 were partly supported. Additionally, the

**Funding:** The author(s) received no specific funding for this work.

organizational resource variable of offering volunteering opportunities was associated with lower burnout levels and moderated the relationship between work-to-family conflict and burnout, thus partially supporting hypotheses H6 and H7.

## Conclusion

The findings enhance the understanding of burnout during prolonged crises like war, highlighting the importance of maintaining routine as a key resource for order and control. They also emphasize the role of organizational volunteering in preventing burnout. The theoretical and practical implications are discussed.

## Introduction

Occupational burnout is a result of ongoing coping with work-related stressors and poses a significant risk to the physical and mental health of employees [1]. It has been defined in the World Health Organization's ICD-11 as a syndrome resulting from "chronic workplace stress that has not been successfully managed" [2]. Burnout manifests as feelings of energy depletion or exhaustion, increased mental distance from one's job, and reduced professional efficacy [3, 4].

The widespread nature and severe consequences of burnout have led professionals and researchers to develop relevant prevention and intervention methods [5]. This effort has recently led to the publication of a new standard by the International Organization for Standardization (ISO 45003). This standard directs organizational responsibility toward promoting mental health at work, preventing and controlling psychosocial risk factors, thereby shifting the focus from the individual, which dominated previous decades, to the organization as a framework for prevention and intervention.

Most research on the causes, manifestations, and consequences of burnout relates to routine life. Occupational burnout during exceptional periods of acute routine disruption, such as during wartime, has been studied to a limited extent, highlighting a gap in the literature. The literature on wartime situations generally focuses on emotional responses and the emergence of mental health issues; however, less considering the psychological reactions from a unique occupational perspective in this context. Therefore, understanding the factors and characteristics of burnout development during exceptional periods can contribute to the prevention of long-term psychological and functional damage documented in the literature. Specifically, living during wartime and terror attacks is accompanied by harsh conditions of risk, uncertainty, and extreme suffering with a general loss of resources, which may accelerate burnout processes in the occupational context as well as having long-term impacts on mental health [6–15].

Elhadi et al [16] examined healthcare workers in Libya during the civil war and the COVID-19 period. The findings indicate high levels of anxiety and depression; 67% of workers reported significant burnout (primarily emotional exhaustion), a high finding compared to the average for this sector in other periods and countries. Tsybuliak et al [17] reported significant burnout among academic staff during the war in Ukraine, particularly in emotional exhaustion (61% of women and 48% of men experienced high levels of emotional exhaustion) and impaired related academic occupational functioning.

Research into the psychological effects of terrorism and war on broad populations is crucial for identifying unique mental health needs following such events. These insights inform the development of preventive strategies and intervention measures tailored to those affected. This study aimed to address a significant gap in literature by exploring burnout among employees

during wartime. Specifically, it examined how daily functioning, personal experiences and organizational resources contributed to burnout levels a month after the war began in Israel. This period profoundly disrupted daily life nationwide, including nationwide alerts and missile strikes, injuries and fatalities, kidnapping incidents, evacuation of communities, and interruptions in educational and employment activities. By focusing on stress, routine disruptions, sleep disturbances, work-family conflicts, and organizational resources as a moderating factor, the research identified key predictors of occupational burnout. These findings highlight the importance of addressing both individual and organizational dimensions in mitigating burnout during crises.

## Burnout: Definition, causes, effects and theoretical background

Burnout is defined as an occupational syndrome characterized by chronic exhaustion, emotional detachment, and cynicism towards one's job, along with reduced professional efficacy [3]. It involves the depletion of energetic resources, chronic fatigue, emotional distancing from work, developing negative attitudes towards the nature of the job and colleagues, and a decline in a sense of capability and achievement [4].

The consequences of burnout are significant and widespread, affecting both individuals and organizations. These include the development of severe physiological and psychological issues such as diabetes, cardiovascular diseases, headaches, depression, anxiety, sleep disorders, and alcohol consumption [18, 19]. There are also negative impacts on organizational job performance, including absenteeism, lack of motivation, reduced productivity, human errors, and employee turnover [20–24]. These phenomena also have a direct impact on economic costs [25].

The Job Demands-Resources (JD-R) model [26] conceptualizes stress and burnout as a response to the intensity of demands and resources present at work. Prolonged exposure to demanding requirements, such as workload, time pressure, and conflicting demands, creates a continual need for physical, emotional, and cognitive effort, leading to resource depletion and the development of burnout [27–29]. Conversely, the presence of job-organizational resources can moderate the relationship between job demands and burnout. For example, Bakker et al [30] demonstrated that workload, demands, and work-home conflict did not lead to increased burnout levels when employees experienced autonomy, received feedback, had social support, and had positive relationships with their direct supervisors. In addition, co-worker support at work was associated with lower levels of burnout during periods of terrorist events in Israel [14]. According to the JD-R model, resources weaken the relationship between demands and burnout by facilitating healthy and effective coping with job demands [31, 32].

The Conservation of Resources theory (COR) posits that individuals are motivated by a desire to obtain, build, and protect resources they value, like property, sense of safety, employment or self-efficacy. Burnout, according to COR theory, is a state of ongoing resource depletion. This state can lead to a spiraling loss cycle, where initial resource loss leads to further losses, increasing stress and making it more challenging to recover and rebuild resources [33, 34]. In this process, the impact of resource loss can spill over into different life domains, leading to chronic burnout [35].

COR theory has been particularly effective in understanding psychological responses to resource losses during extreme crises affecting large populations. This theory explains variations in the impact of traumatic stress following significant events such as natural disasters, terrorist attacks, pandemics, and prolonged security conflicts [36–42]. Research has shown that the level of resource loss, especially personal and social resources, is directly related to the level of psychological distress following traumatic events. For example, there is a documented relationship between resource loss and psychological distress following Hurricane Georges [43],

the September 11 attacks in New York [44], prolonged exposure to terrorism [38, 45], and fear of terrorist events [14].

Living during wartime involves significant loss of security, stability, social and economic resources, as well as trust and hope. Studies have shown that the extent to which an individual experiences functional and psychological resource loss during periods of terrorism and war is significantly related to post-traumatic symptoms [46]. Living under the fear of terror, with ongoing concern for essential life resources such as family and property, create a high level of stress that can lead to burnout [14]. Resource loss has also been found to be a significant factor in emotional distress during and after the COVID-19 pandemic [47].

Alongside the well-documented relevance of the Conservation of Resources theory in explaining negative psychological phenomena of stress during periods of extreme crises, there is a need for more research regarding the mechanisms through which resource depletion occurs and leads to resource loss in these periods [14]. This study examines the associations between various resources that may be disrupted during wartime, including daily routines, work-to-family conflict, family-to-work conflict, and sleep problems, as mechanisms through which burnout develops during extreme periods of war.

## Disruptions in routine, sleep and work-family conflict as resources loss during wartime

Routine—daily schedules, temporal continuity, and repetitive daily actions—constitutes a central functional and psychological resource in life [48]. Routine provides structure, framework, and rhythm to activities, allowing synchronization across different life domains and generating psychobiological cues that guide the timing of actions (bedtime, meals, work, leisure time with children and more). Disruptions in routine have been linked in literature to anxiety, depression, and post-traumatic stress disorder (PTSD) [49–51]. Li et al [49] found that the relationship between routine disruptions and emotional distress is mediated by reductions in self-efficacy and coping resources.

Prolonged stress and traumatic events create contexts that restrict individuals from performing routine tasks—both objectively, such as the cessation of work, education systems, and economic activities, and personally, focusing on the stressor and the emotional-behavioral response to it rather than on daily routine activities. The inability to maintain a routine (reduced adherence to eating, sleeping, leisure, social, and occupational activities) has been documented in research as a common phenomenon among populations under prolonged stress and is significantly associated with psychological distress [49–52]. Studies during the COVID-19 pandemic found that disruptions in routine (sleep, exercise, nutrition, work habits) were linked to poor emotional states and blurred boundaries between day and night, leisure, and work. Research among healthcare workers during the pandemic found that routine disruptions were one of the predictors of both emotional distress and burnout [53].

As routine serves as a foundational resource, balancing central life roles like work and family may also serve as a resource. Family and work are two fundamental roles in adult life, each demanding numerous tasks, time commitments, and simultaneous functional demands. This can pose significant challenges in achieving balance, especially during stressful periods such as wartime. Consequently, the balance between these roles may be disrupted, leading to conflict.

Work family conflict refers to an inter-role conflict where pressures from the work and family domains are mutually incompatible [54]. The direction of this conflict can be defined as work interfering with family (work-to-family conflict) or family interfering with work (family-to-work conflict) [55]. Work family conflict is a well-established finding in burnout literature, as higher levels of work family conflict are consistently linked to increased burnout [56].

Based on COR theory [57], work and family domains can be viewed as reservoirs of resources, where a potential or actual threat or loss in one domain affects the basic state of the other [58]. Various studies have emphasized that global and significant crisis events, such as the COVID-19 pandemic and the war in Ukraine, have further blurred the boundaries between work and non-work roles [59–62], changing how individuals navigate between these roles. However, research regarding work family conflict and burnout during wartime is missing.

Sleep represents the 'non-wakeful' component of routine and has been directly associated with other daily disruptions and measures of emotional distress [63, 64]. Sleep disturbances— difficulty falling asleep or maintaining sleep [65]—are prevalent during prolonged exposure to traumatic events [66]. Stressful experiences contribute to autonomic arousal and emotional distress, which are linked to sleep difficulties [67]. Sleep problems, in and of themselves, reduce physical energy and mental resources, contributing to the development of burnout [68, 69].

## Organizational means as resource enhancement during wartime

According to the foundational assumptions of the JD-R model, organizational resources are aspects of work that can be functional in achieving work goals, reducing demands and their physiological and psychological costs, and fostering personal growth and development. Examples of organizational resources include support from others and receiving feedback on work, as these resources are negatively related to burnout [28]. Organizational resources are associated with motivational processes, leading to positive outcomes such as organizational commitment, intentions to continue a career within the organization, extra-role contributions, safety behavior, and more. The presence of organizational resources can fulfill psychological needs and mitigate the impact of work demands on burnout. Organizational resources weaken the associations between demands and burnout by facilitating efficient and healthy coping with work demands [31, 32].

In addition to interventions aimed at strengthening personal resources, the literature documents various interventions to enhance organizational resources. Both types of interventions generally lead to improvements in preventing burnout, but the improvement is often relatively small or sometimes negligible [70]. One explanation for this is the use of intervention measures without prior measurement and research.

Among organizational resources, a resource that may hold significant value during wartime is flexibility in working hours. Flexibility has been widely recognized in the literature as a crucial resource for employees and as a buffer against burnout when faced with high demands [71, 72]. Work flexibility is defined as "the ability of employees to choose and influence where, when, and for how long they invest in work tasks" [73]. It is hypothesized that employees with flexible work schedules can better balance their work-home boundaries and experience greater autonomy and control in this context. The current study aims to contribute to the existing literature regarding the value of flexibility as an organizational resource during wartime, hypothesizing that flexibility provided by the organization will enable employees to better manage new and extreme demands at home and work during such periods.

Another significant organizational resource examined in this study is providing access to volunteering opportunities for employees. Volunteering is an activity that enables employees to experience an active, meaningful, and contributive approach, leading to better well being [74, 75]. Volunteering has also been shown to foster strong connections with the broader community which serves as a critical vital force during crises by reinforcing shared identity and resilience [76, 77]. During the COVID-19 pandemic, volunteers were found to experience a higher level of community identification and belonging compared to non-volunteers [78].

Against this backdrop, it is hypothesized that offering volunteering opportunities as part of organizational practice will have unique value in enhancing employee experiences.

## Current research

Based on the Conservation of Resources (COR) theory, the current study examined the associations between depletion of employee resources and the experience of occupational burnout during wartime. The researchers first aimed to assess and report the levels of burnout experienced by the sample under wartime conditions. Additionally, the study tested the following hypotheses:

H1. Stress would be positively associated with occupational burnout.

H2. Work-to-family conflict would be positively associated with occupational burnout.

H3. Family-to-work conflict would be positively associated with occupational burnout.

H4. Sleep disruption would be positively associated with occupational burnout.

H5. Routine disruption would be positively associated with occupational burnout.

In addition, the study also evaluated whether organizational strategies designed to enhance resources can help mitigate burnout during wartime. Therefore, it was hypothesized: H6. Organizational resources- flexibility and volunteering opportunities- would be negatively associated with burnout.

Furthermore, given that work-family conflict was a well-established predictor of burnout in the existing literature, even under normal conditions, it was hypothesized that: H7. The organizational resources- flexibility and volunteering opportunities- would moderate the relationship between work-family conflict and burnout during wartime.

## Materials and methods

### Participants

The study included 374 participants consisting of 261 women, 112 men, and one unreported, with ages ranging from 28 to 60. The inclusion criteria were individuals employed full-time or part-time who were not in military service and had not been evacuated from their homes due to the war. The study was conducted a month after the war began in Israel (additional information about the war is detailed in the Recruitment and Procedure section), therefore participants were asked whether they or their family members or close friends had been physically injured directly during this month, by an act of terror or war. Only 4 participants (1.1%) reported they were physically injured, as 37 participants (17.1%) reported physical injuries of their friends or relations. The sociodemographic data and information concerning the war are presented in Table 1.

### Recruitment and procedure

The war began on October 7, 2023, and has been ongoing for over a year. The study was conducted between November 7 and November 19, 2023, one month after the war began. Ethical approval for this study was obtained from the Institutional Review Board (IRB) of The Academic College of Tel-Aviv Yaffo. Participants were recruited using a snowball sampling technique through various social media networks (e.g., Facebook, WhatsApp). Questionnaires were administered via the Qualtrics online platform (www.qualtrics.com). Participants were invited to participate in a survey on coping with the war in Israel. They were assured that their

**Table 1. Sociodemographic data and organization variables.**

| Variables | n (%) | Correlation with occupational Burnout |
|---|---|---|
| Gender | | **.18**** |
| Women | 261 (69.8%) | |
| Men | 112 (29.9%) | |
| Unreported gender | 1 (0.3%) | |
| Age | | n.s |
| 28–34 | 152 (40.6%) | |
| 35–40 | 75 (20.1%) | |
| 41–45 | 53 (14.2%) | |
| 46–50 | 33 (8.8%) | |
| 51–55 | 45 (12%) | |
| 56–60 | 16 (4.3) | |
| Family Status | | n.s |
| Single\ Divorced\ Widow | 103 (27.5%) | |
| Married\ co-habiting | 271 (72.5%) | |
| Higher education | | n.s |
| Bachelor's Degree | 149 (39.8%) | |
| Master's Degree | 146 (39%) | |
| Doctorate Degree | 19 (5.1%) | |
| High School / Professional Diploma | 56 (15%) | |
| Missing | 4 (1.1%) | |
| Physically injured (me) | | n.s |
| Yes | 4 (1.1%) | |
| No | 370 (98.9%) | |
| Physically injured (close others) | | n.s |
| Yes | 64 (17.1%) | |
| No | 310 (82.9%) | |
| Organization resources variables | M (SD) | |
| Flexibility in work hours | 3.70 (1.16) | -.02 |
| Opportunities for volunteering | 3.15 (1.40) | **-.12*** |

*Note*: N = 374.

*$p < .05$

**$p < .01$.

Full details regarding the nature of the organization-related questions can be found in the 'Questions regarding organizational resources' section of the Method.

responses would remain anonymous, and participation was voluntary. All participants provided informed consent before proceeding with the survey. They were informed that they could discontinue participation and withdraw from the study at any point. The researchers' contact information was provided.

## Measures

**Sociodemographic questionnaires.** Included gender, age, family status, and education level. Additionally, participants were asked about physical injuries due to the war.

**Questions regarding organization resources.** Two questions were posed regarding the participants' work organization. During the war, my organization provided: 1. Flexibility in work hours and formats. 2. Opportunities for volunteering and contributing to the community

during the war (such as assisting in agriculture, preparing food, providing mental support for soldiers and families, and more).

**Occupational burnout.**   Measured by the Hebrew version of the Shirom-Melamed Burn-out Measure (SMBM), a 14-item measure assessing job burnout. Participants were asked to report how often they had experienced physical fatigue, cognitive weariness, and emotional exhaustion in the last month at work. All items were scored on a 7-point scale, ranging from 1 (almost never) to 7 (almost always). The mean score across the 14 items was calculated [79]. In the present study, the SMBM demonstrated strong internal consistency, with a Cronbach's alpha coefficient of .94.

**Stress.**   Measured by the Hebrew version of the Depression, Anxiety, Stress Scales (DASS-21), a 21-item self-report questionnaire designed to evaluate symptoms of depression, anxiety, and stress experienced over the past week. Each subscale consists of seven items. In the current study the researchers used the stress scale. Respondents rated their experiences on a four-point Likert scale ranging from 1 (did not apply to me at all) to 4 (applied to me very much), with scores calculated as the sum of responses for each item in the subscale [80, 81]. In the present study, the stress scale demonstrated strong internal consistency, with a Cronbach's alpha coefficient of .90.

**Work-family conflict.**   Assessed using an eight-item measure developed for the MIDUS study by Allen et al [82]. Four items measured work-to-family conflict, while four measured family-to-work conflict. Respondents rated themselves on a five-point Likert scale ranging from 1 (Never) to 5 (All of the time). The questionnaire was translated into Hebrew for this study using forward-backward translation by the researchers [83]. In the current study, both work-to-family conflict and family-to-work conflict demonstrated strong internal consistency, with Cronbach's alpha coefficients of .82 and .76, respectively.

**Routine disruptions.**   Assessed using the Sustainability of Living Inventory (SOLI). This self-report instrument measured perceived disruptions in the regularity of daily routines. In the present study, the researchers used the subscales for eating, exercising, social activities, and work/study involvement [48]. The questionnaire was translated into Hebrew for this study using forward-backward translation by the researchers [83]. The SOLI scale demonstrated strong internal consistency in this study, with a Cronbach's alpha coefficient of .82.

**Sleep disruptions.**   Assessed using the Insomnia Severity Index (ISI), a 7-item self-report questionnaire assessing the nature and symptoms of the participants' sleep problems during the last month. The dimensions evaluated were the severity of sleep onset, sleep maintenance, early morning awakening problems, sleep dissatisfaction, interference of sleep difficulties with daytime functioning, noticeability of sleep problems by others, and distress caused by sleep difficulties. A 5-point Likert scale was used to rate each item from 0 (no problem) to 4 (very severe problem), yielding a total score ranging from 0 to 28 [84]. In the present study, the stress scale demonstrated strong internal consistency, with a Cronbach's alpha coefficient of .87.

### Statistical analysis

To determine the appropriate sample size for the study, a power analysis was conducted using the G*Power program. Based on an expected medium effect size, a power of 0.95, and an alpha probability error of 5% for six predictors (including covariates), the researchers calculated that a minimum of 146 participants would be required. Data were presented as means and standard deviations or as counts and percentages.

Correlations among the study variables were evaluated using the Pearson correlation coefficient. Hierarchical linear regression was utilized to model the outcome variables, occupational burnout. In the first step, gender and stress were included as they are considered as covariate

and the mental state considering the war situation. In the second step, work-to-family conflict, family-to-work conflicts, sleep disruption and routine disruption were added. Results were reported in terms of estimated standardized coefficients (β), unstandardized coefficients (B), standard errors (SE), and associated p-values. Adjusted explained variance ($R^2$) and associated F-values were also reported for each step. A significance level of p < .05 was considered statistically significant. To test the moderation hypothesis, the researchers utilized Process Model 1. Data analysis was conducted using the statistical software package SPSS 28.0 (SPSS Inc., Chicago, IL) and PROCESS Version 4.2 [85].

## Results

The mean level of occupational burnout among participants during wartime was notable (M = 3.95, SD = 1.15). Table 1 presents the sample demographics characteristics and the correlations between them and the outcome variable occupational burnout. As shown in Table 1, significant differences were observed in occupational burnout levels between women and men, with women exhibited higher levels of occupational burnout. Consequently, gender was included in the regression analysis. Results showed no other significant differences between the demographic variables and the outcome variable.

To test the research hypotheses (H1 to H5), Pearson correlation analyses were conducted, followed by hierarchical regression as a post hoc analysis to further examine the relative contributions of each variable to burnout during wartime. Table 2 presents the correlations between the study variables. The factors significantly correlated with occupational burnout were stress (r = .63**), work-to-family conflict (r = .47**), sleep disruption (r = .44**), family-to-work conflict (r = .42**), and routine disruption (r = -.35**). The regression results are presented in Table 3, with the model being significant at each step. In step 2, the strongest predictors of occupational burnout were stress (β = .43**), work-to-family conflict (β = .15**), routine disruption (β = -.14**), and sleep disruption (β = .09**). Gender (β = .08) and family-to-work conflict (β = .07) did not significantly contribute to occupational burnout during the war. In step 2, the model explained 48% of the variance in occupational burnout. Thus, hypotheses H1, H2, H4, and H5 were supported, while H3 was not.

To examine hypothesis H6, the researchers analyzed the associations between organizational resources—flexibility and opportunities for volunteering in the workplace—and occupational burnout using Pearson correlations (Table 1). Workplace flexibility was not significantly associated with burnout. However, opportunities for volunteering were significantly associated with burnout. Consequently, only volunteering opportunities were analyzed as a moderator in the relationship between work-to-family conflict and burnout (H7). At higher levels of

**Table 2. Pearson's correlation coefficients between the study variables.**

| | M (SD) | 1. | 2. | 3. | 4. | 5. | 6. |
|---|---|---|---|---|---|---|---|
| 1. occupational Burnout | 3.95 (1.15) | - | .63** | .47** | .42** | .44** | -.35** |
| 2. Stress | 8.38 (4.80) | | - | .47** | .46** | .50** | -.28** |
| 3. Work to family conflict | 2.87 (0.83) | | | - | .55** | .33** | -.24** |
| 4. Family to work conflict | 2.60 (0.77) | | | | - | .39** | -.22** |
| 5. Sleep disruption | 13.18 (6.32) | | | | | - | -.32** |
| 6. Routine disruption | 2.45 (0.52) | | | | | | - |

*Note*:

*p < .05

**p < .01. stress and sleep disruption scores are presented as average totals per respondent.

**Table 3. Hierarchical regression analysis with <u>occupational burnout</u> as the outcome variable.**

| Predictors | β | B | SE (B) | |
|---|---|---|---|---|
| *Step 1-* | | | | |
| (constant) | | 2.27 | .18 | $R^2 = .41$ |
| Gender | **.10*** | .26 | .10 | $F(2, 371) = 130.02^{**}$ |
| Stress | **.62**** | .15 | .01 | |
| *Step 2-* | | | | |
| (constant) | | 2.42 | .36 | |
| Gender | .08 | .19 | .10 | |
| Stress | **.43**** | .10 | .01 | |
| Work-to-family conflict | **.15**** | .20 | .06 | $\Delta R^2 = .07$ |
| Family-to-work conflict | .07 | .10 | .07 | $R^2 = .48$ |
| Sleep disruption | **.09*** | .02 | .008 | $F(6, 367) = 56.40^{**}$ |
| Routine disruption | **-.14**** | -.31 | .09 | |

Note: N = 374.

*$p < .05$

**$p < .01$.

volunteering opportunities, the association between work-to-family conflict and burnout became insignificant. Thus, hypotheses H6 and H7 were partially supported. The results of the moderation model are presented in Table 4 and Fig 1.

## Discussion

Burnout, an occupational psychological phenomenon, has not been sufficiently researched in the context of a functioning population during prolonged stress and crisis periods. Therefore, the current research aimed to measure occupational burnout during wartime in Israel. The results showed a high average burnout score (M = 3.95, SD = 1.15). This finding can be compared to a national study by Egosi and Peredo [86], which reported an average burnout score of 3.01 (SD = 1.31) before the war and a mean level of 3.55 (SD = 1.41) two months after the war began in Israel. This research makes a unique contribution to the field by not only confirming the well-documented effects of war on mental health in populations, such as the onset of depression, anxiety, and trauma [16, 29, 46] but also by emphasizing occupational burnout

**Table 4. Regression analysis testing the moderation hypothesis with occupational burnout as the outcome variable (N = 374).**

| Predictors | B | S.E | P |
|---|---|---|---|
| Work-to-family conflict | **0.53**** | 0.13 | .001 |
| Organization volunteer | 0.20 | 0.11 | .08 |
| Work-to-family conflict * Volunteer | **-0.76*** | 0.37 | .04 |
| Stress | **.13**** | .01 | .000 |

Note: work-to-family conflict as the independent variable, volunteering as the moderating variable, and stress as a covariate.

F(4, 369) = 73.90, $R^2 = .44$

*p < 0.05

**p < 0.01

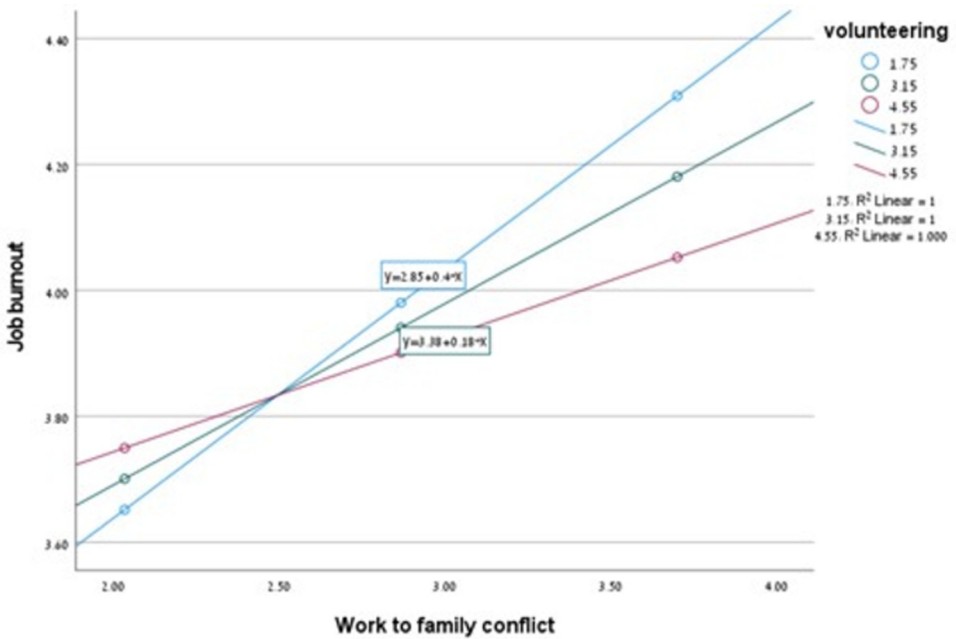

**Fig 1. Simple slopes of the moderation model.** Note: Regression lines of the association between increase in work-to-family conflict and occupational burnout are shown for M - SD (1.75), M (3.15) and M + SD (4.55) values of organizational volunteer opportunities.

under wartime conditions. These findings are particularly significant given the limited literature on burnout during wartime or prolonged national crises.

The researchers aimed to measure disruptions as resource loss among employees and their associations with burnout during wartime. It was found that the strongest predictors of occupational burnout were stress ($\beta$ = .43**), work-to-family conflict ($\beta$ = .15**), routine disruption ($\beta$ = -.14**), and sleep disruption ($\beta$ = .09**). Gender ($\beta$ = .08) and family-to-work conflict ($\beta$ = .07) did not significantly contribute to occupational burnout during the war. As the Conservation of Resources [57] theory suggested, war creates resource loss and a persistent threat of resource depletion. This study addresses unique resource loss during wartime—loss of the balance between work and family roles, loss of structure and function of daily routines: social, occupational, and nutritional routines, and disrupted sleep. All were found to contribute to higher levels of burnout.

The findings also highlight the significant role of routine disruption in understanding burnout during wartime. These results align with the Drive to Thrive Theory (DTT) [87], which introduced the concept of "fabrics" to represent daily routines. According to DTT, personal, social, and material resources are sustained through routine experiences. Fabrics include biological, psychological, community, and social activities that establish a structured life pattern. Hou et al. suggested that resilience to prolonged stress is defined by how well people maintain these fabrics and daily life structures. Adapting to traumatic events or extended stress can disrupt daily routines, as individuals prioritize attention to stressors and overall distress, making it challenging to maintain a regular schedule and daily habits. In this study, a month following the beginning of the war, routine disruptions contributed significantly to burnout.

The results aligned with previous studies that underscored the association between work-to-family conflict and occupational burnout [56]. During the war, this sample demonstrated that work-to-family conflict was associated with higher burnout, whereas family-to-work

conflict showed no significant association. This finding emphasized that the conflict is a dynamic phenomenon that changes based on circumstances and situations [88, 89]. It is plausible that employees prioritized their family obligations over work during wartime, focusing on caring for loved ones, leading them to view their work role as a more significant source of conflict.

Lastly, the regression model revealed a positive association between sleep disruptions and burnout. This finding aligned with previous studies, emphasizing that sleep problems reduce physical energy and mental resources, thereby contributing to burnout [68, 69].

Gender was directly associated with burnout in this sample, with women exhibited higher burnout levels than men. The results are consistent with a study on the war in Ukraine, which also found higher levels of burnout among women [17]. However, in the regression model, gender did not significantly contribute to burnout. It appeared that the experience of stress and disruptions played a more significant role than gender in explaining burnout.

Based on COR theory and an organizational perspective, the researchers investigated whether special types of organizational resources could significantly contribute to reducing burnout. It was found that participants who reported that their organizations provided more volunteering opportunities, exhibited lower levels of burnout. This is supported by a broad literature emphasizing the importance of organizational resources in buffering burnout, even under high-demand conditions [30, 31]. Increasing resource accessibility, guided by Conservation of Resources theory principles, holds particular value in situations of resource depletion processes [30].

Interestingly, and contrary to the hypothesis, flexibility in work hours and formats was not associated with burnout. The findings in the literature regarding flexibility are inconclusive. Contrary to their expectations, Penpek et al [90] found that among employees with health issues during the COVID-19 period, flexibility did not serve as a moderating factor for burnout. Some meta-analysis reviews indicate a relatively low contribution of flexibility in addressing work-home conflict [71, 91], suggesting that the impact of flexibility is context-dependent. Our findings indicated that during times of war, flexibility may not act as a resource; it might even have the opposite effect. In the uncertainties of routine and daily life, there may be a need to restore structure and order rather than prioritize flexibility.

Volunteer opportunities, provided by the organization, moderated the association between work-to-family conflict and occupational burnout; with higher volunteer opportunities, the association between work-to-family conflict and burnout became insignificant. In other words, among employees who perceived their organizations as offering numerous volunteering opportunities during wartime, no significant relationship was found between work-to-family conflict and burnout. This supported the JD-R model, emphasizing that resources reduced the associations between demands and burnout, as organizational job resources moderated the impact of demands on stress development by creating motivation to cope positively with those demands [30]. However, it is important to identify the relevant contribution of different resources, across various periods and contexts. Burgess et al [70] presented a comprehensive review of the impact of various organizational intervention measures in promoting mental health at work, noting the lack of positive effects of many of them. They concluded that focusing on precise, interview-based intervention methods was crucial.

In this case, volunteer opportunities during wartime may have provided added value, meaning, and a sense of vitality and relevance in the context of the larger situation (war). By doing so, organizational volunteers helped reduce the tension individuals felt between their work and family functions, and its association with burnout. Volunteering enhanced self-efficacy and active capability, which research has identified as significant mediating factors between routine disruptions and emotional distress during prolonged crises.

The findings emphasized that organizations can serve as stabilizing agents during crises, as resources provided by organizations can help mitigate work-related burnout and emotional distress during external crises. Hobfoll et al [92] identified five critical components for intervention following crises, disasters, and prolonged trauma among the general population. One of these components was developing a sense of personal and collective self-efficacy. When exposed to trauma and prolonged crises, individuals often lose their sense of competence in the face of overwhelming and uncontrollable events. This feeling, which develops directly in response to initial events, can be generalized into a pervasive sense of 'cannot do'. A primary goal of preventive and therapeutic intervention in this context is to restore the sense of competence and capability, mainly through active engagement. The ability to restore and maintain routine life, actively cope with difficulties, and engage in meaningful activities related to the overall crisis is crucial for therapeutic guidance and organizational support. In sum, understanding burnout during wartime is crucial for preventing long-term psychological and functional damage. Harsh conditions of risk, uncertainty, and suffering can deplete resources and accelerate burnout in the workplace, ultimately having lasting effects on mental health. The workplace, a central aspect of a person's life, has the potential and the responsibility to act optimally to promote the mental health of individuals.

## Limitations and future research

Despite its importance, the research has limitations. Firstly, it was conducted at a single time point, which restricts the ability to draw causal inferences. Occupational burnout was measured after the beginning of the war, so the findings reflect employees' perceptions of burnout only during the data collection period. A long-term study could provide a more accurate understanding of burnout levels over time during wartime, while also allowing for a broader examination of other mental health variables, such as PTSD.

In addition, longitudinal studies that assess measures of distress and adaptation years after the end of a war can illuminate the persistence of burnout conditions and the impact of real-time access to organizational resources on long-term adaptation, resilience, and quality of life.

Another limitation is focusing only on routine disruptions in the burnout process during wartime. It is crucial to broaden the examination of these disruptions within the context of broader resource loss and among specific populations, such as individuals with physical and mental injuries or evacuees from high-risk areas. Given that the study utilized a snowball sampling method and focused on a population not directly affected by the war, expanding the study's scope could be beneficial. This expansion could help identify population-specific risk factors and allow for the customization of intervention and prevention measures accordingly. It would be valuable to examine different job types and work characteristics and their associations with burnout during wartime. Furthermore, exploring the impact of interventions specifically targeting the maintenance of routine during wartime—across various dimensions—on short-term and long-term measures of adaptation and quality of life would also be valuable.

## Conclusions

A routine is a structure of order, organization, control, and regular habits. It delineates boundaries between life domains, activity rhythms, rest, and sleep. The loss of this structure represents a crucial depletion of essential resources for vital functioning, especially during times of prolonged crisis and continuous stress. The current research indicated that the loss of routines, work-family balance, and sleep were associated with higher levels of burnout. These findings underscored the importance of developing interventions to restore and maintain routines, such as strategies to reduce stress, ensure continuity of work and daily activities (including

physical activity, social connections, and nutrition), and emphasize the importance of sleep. Additionally, helping employees balance their work and family responsibilities is crucial, especially among populations experiencing prolonged crises like war. The significance of organizational resources in preventing burnout is another critical finding in the current research, highlighting that providing accessible resources—specially tailored resources aimed at enhancing self-efficacy and practical coping with meaning and significance—are effective mechanisms for maintaining the mental and functional quality of life during extended crises. Future research should aim to better identify the effects of various organizational measures in response to different types of prolonged crises, such as war, economic instability, pandemics, and natural disasters. Additionally, examining the distinct impact of organizational resources measures on diverse employee groups—such as men and women, different age groups, and single versus family-status employees—would be valuable. Expanding research in these two directions could provide practical guidance for tailoring intervention measures more precisely to specific contexts.

## Author Contributions

**Conceptualization:** Maor Kalfon Hakhmigari, Irene Diamant.

**Formal analysis:** Maor Kalfon Hakhmigari.

**Investigation:** Maor Kalfon Hakhmigari, Irene Diamant.

**Methodology:** Maor Kalfon Hakhmigari.

**Supervision:** Irene Diamant.

**Writing – original draft:** Maor Kalfon Hakhmigari, Irene Diamant.

**Writing – review & editing:** Maor Kalfon Hakhmigari, Irene Diamant.

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
