## [Decision Letter · Decision Letter 0]

16 Sep 2024

PONE-D-24-34427Occupational burnout during war: The role of disruptions in routine, sleep, work-family conflict, and organizational support as a moderatorPLOS ONE

Dear Dr. Kalfon Hakhmigari,

Thank you for submitting your manuscript to PLOS ONE. After careful consideration, we feel that it has merit but does not fully meet PLOS ONE’s publication criteria as it currently stands. Therefore, we invite you to submit a revised version of the manuscript that addresses the points raised during the review process. Please submit your revised manuscript by Oct 31 2024 11:59PM. If you will need more time than this to complete your revisions, please reply to this message or contact the journal office at plosone@plos.org. Please include the following items when submitting your revised manuscript:A rebuttal letter that responds to each point raised by the academic editor and reviewer(s). You should upload this letter as a separate file labeled 'Response to Reviewers'.A marked-up copy of your manuscript that highlights changes made to the original version. You should upload this as a separate file labeled 'Revised Manuscript with Track Changes'.An unmarked version of your revised paper without tracked changes. You should upload this as a separate file labeled 'Manuscript'.

We look forward to receiving your revised manuscript.

Kind regards,

Supaprawat Siripipatthanakul, Ph.D.,DBA, MS. (Management), DDS. etc.

Academic Editor

PLOS ONE

Journal Requirements: When submitting your revision, we need you to address these additional requirements. 1. Please ensure that your manuscript meets PLOS ONE's style requirements, including those for file naming. The PLOS ONE style templates can be found at https://journals.plos.org/plosone/s/file?id=wjVg/PLOSOne_formatting_sample_main_body.pdf and https://journals.plos.org/plosone/s/file?id=ba62/PLOSOne_formatting_sample_title_authors_affiliations.pdf 2. Please ensure that you include a title page within your main document. You should list all authors and all affiliations as per our author instructions and clearly indicate the corresponding author. 3. Please amend your manuscript to include your abstract after the title page. 4. Please include your full ethics statement in the ‘Methods’ section of your manuscript file. In your statement, please include the full name of the IRB or ethics committee who approved or waived your study, as well as whether or not you obtained informed written or verbal consent. If consent was waived for your study, please include this information in your statement as well. 

Additional Editor Comments:

Please create four tables for each of the two columns. The left column is for reviewers' comments; the other is for the author(s)' revision (one-by-one comment and revision). Kindly carefully read and follow the comments. In case of disagreement (but please avoid this), you may explain why you argue and do not intend to follow the comments. Please highlight the revised statement in yellow. The academic editor's decision is revisions are required, but there is a high probability of acceptance after major revisions. Before resubmission, please check for plagiarism and similarity with AI-generated documents below 15%.

Reviewers' comments:

Reviewer's Responses to Questions

**Comments to the Author**

1. Is the manuscript technically sound, and do the data support the conclusions?

Reviewer #1: Partly

Reviewer #2: Yes

Reviewer #3: Yes

Reviewer #4: Partly

2. Has the statistical analysis been performed appropriately and rigorously? 

Reviewer #1: Yes

Reviewer #2: Yes

Reviewer #3: Yes

Reviewer #4: Yes

3. Have the authors made all data underlying the findings in their manuscript fully available?

Reviewer #1: Yes

Reviewer #2: Yes

Reviewer #3: Yes

Reviewer #4: Yes

4. Is the manuscript presented in an intelligible fashion and written in standard English?

Reviewer #1: Yes

Reviewer #2: Yes

Reviewer #3: Yes

Reviewer #4: No

5. Review Comments to the Author

Reviewer #1: Line 37: the main objective of the current research was to address the literature gap regarding

burnout during wartime, which is not present clearly in the research paper.

A major concern is the clarity regarding participant selection, such as how they were recruited and the specific inclusion and exclusion criteria applied.

Line 205: Re-check with data in Table 1. , It is suggested to address One missing participant.

Line 207: It is suggested to specify the type of employees involved in this research, including their job functions and tasks, as different jobs and tasks result in different levels of stress.

Line 209:It is suggested to specify the timeframe, such as having sustained a physical injury within 2 months or within 1 year and specify the severity of the physical injuries being counted.

In Table 1 -

Gender : It is suggested to re-check one Adding participant (Line 205).

Age : Is there any information on the average age, minimum age, and maximum age of the participants?

This is an unequal in each age group; 28-34 (=8), 35-40(=5)

Higher Education : It should be categorized by educational level: bachelor's degree, master's degree, and doctorate degree.

Organization Variable : This part should be clarified further: What is the Organization Variable? Are these related to Job Characteristics?

Line 222 : What is the inclusion/exclusion criteria for this study ?

Line 235 : Has the nature of this work changed compared to how it was before the war?

Line 259 : What does MIDUS stand for?

Reviewer #2: According to Table 4, Predictors of Occupational burnout during war are

(1) stress (2) Work to family conflict (3) Family to work Conflict (4) Sleep disruption (5) Routine disruption. And organizatonal support is the moderator. Thus, the recommended topic is

"Stress, Work to Family Conflict, Family to work Conflict, Sleep disruption and Routine disruption Predicting Occupational Burnout: The Moderator of Organizational Support". In the astract participants completed a questionnaire assessing demographic details, and variables should be 1) stress (2) Work to family conflict (3) Family to work Conflict (4) Sleep disruption (5) Routine disruption. And organizational support is the moderator. Please add data analysis in the abstract. Please use researchers instead of we or our. The introduction, literature review, methods, results, discussion, conclusion, limitations, and references are okay.

Reviewer #3: The manuscript is well-structured, technically competent, and contributes to wartime occupational burnout research. The study rigorously tests its hypotheses using hierarchical regression and moderation analysis. Well-reported data support conclusions. To ensure openness and data-sharing compliance, all underlying data are available. The article uses ordinary English to define fundamental ideas and explain theoretical frameworks. Dual publication, research ethics, and publishing ethics seem fine. Overall, the text offers significant insights for crisis employee burnout management. The study followed ethical guidelines and received institutional review board approval, thus there are no ethical problems. The study offers data ethically and with integrity. Future studies should examine burnout's long-term effects and interventions. I advocate publishing this work following minor edits to clarify parts like the practical consequences of the findings. This study advances occupational burnout theory and practice in war-torn areas.

Reviewer #4: • The manuscript covers an important and novel topic by exploring burnout during war conditions with relevant factors. However, this study still lacks several important factors, such as occupation and daily working hours.

• Measuring occupational burnout after the occurrence of war may not be appropriate, as it involves post-traumatic stress disorder. Therefore, the findings may not accurately represent employees’ occupational burnout.

• I would like the author to consider expanding the discussion on why focusing on burnout during war is particularly important and how it could be relevant to policymakers, organizations, and family members.

6. PLOS authors have the option to publish the peer review history of their article (what does this mean?). If published, this will include your full peer review and any attached files.

Reviewer #1: No

Reviewer #2: No

Reviewer #3: No

Reviewer #4: No

---

## [Author Response · Author response to Decision Letter 0]

4 Oct 2024

Thank you for the review. We have implemented the required changes and included a response letter addressing the reviewers' comments.

---

## [Decision Letter · Decision Letter 1]

15 Oct 2024

PONE-D-24-34427R1Occupational burnout during war: The role of stress, disruptions in routine, sleep, work-family conflict, and organizational support as a moderatorPLOS ONE

Dear Dr. Kalfon Hakhmigari,

Thank you for submitting your manuscript to PLOS ONE. After careful consideration, we feel that it has merit but does not fully meet PLOS ONE’s publication criteria as it currently stands. Therefore, we invite you to submit a revised version of the manuscript that addresses the points raised during the review process.

We look forward to receiving your revised manuscript.

Kind regards,

Supaprawat Siripipatthanakul, Ph.D.

Academic Editor

PLOS ONE

**Additional Editor Comments:**

Additional comments from Academic Editor.

(1) Kindly follow reviewer 1's comments.

(2) In the abstract, please add that hierarchical linear regression and Pearson Correlation were utilized to model the outcome variable of organizational burnout.

(3) In The last paragraph of the introduction, please add: Thus, this study identified the predictors of occupational burnout based on stress, disruptions in routine, sleep, work-family conflict, and organizational support as a moderator (Line 48).

(4) Kindly check capital letters, such as occupational Burnout and Process Model 1 (in the abstract); it should be

"occupational burnout and Process Model 1", or "Occupational Burnout and Process Model 1".

(5) Table 2. Please add: The significant correlated factors of occupational burnout are stress (r=.63**), work-to-family conflict (.47**), sleep disruption (r=.44*), family fo work conflict (r=.42**), and routine disruption (r=-.322**), respectively.

(6) Table 3. Step 2: Please add that the most influencing factors of occupational burnout are stress (β=.43**), work to family (β=.15**), routine disruption (β=-.14**), sleep disruption (β=0.98**), and family fo work (β=0.07), at 2= .48 (occupational burnout explanation power is 48%). Kindly check if gender is added, which may reflect different results with Pearson Correlation in Table 2 in important priority.

(7) Table 4. Please use Predictors instead of Predictor.

(8) Discussion and conclusion should summarize the findings, such as,

The significant correlated factors of occupational burnout are stress (r=.63**), work-to-family conflict (.47**), sleep disruption (r=.44*), family fo work conflict (r=.42**), and routine disruption (r=-.322**), respectively.

Comparison with regression: the most influencing factors of occupational burnout are stress (β=.43**), work to family (β=.15**), routine disruption (β=-.14**), sleep disruption (β=0.98**), and family fo work (β=0.07), at 2= .48 (occupational burnout explanation power is 48%).

(9) Please ensure that the conclusion is made on how strategic planners could improve organizational burnout based on stress, work-to-family conflict, sleep disruption, family-of-work conflict, and routine disruption. (A summarized table of hypotheses is suggested in the result section, if applicable).

(10) Please ensure that volunteering or organizational support is the moderator. Please use only one term or mention anywhere if these two terms are interchangeable.

(11) Limitations should be added about the time, such as measuring occupational burnout after the occurrence of war; the findings represent employees' occupational burnout perceptions during the data collection period.

Reviewers' comments:

Reviewer's Responses to Questions

**Comments to the Author**

1. If the authors have adequately addressed your comments raised in a previous round of review and you feel that this manuscript is now acceptable for publication, you may indicate that here to bypass the “Comments to the Author” section, enter your conflict of interest statement in the “Confidential to Editor” section, and submit your "Accept" recommendation.

Reviewer #1: (No Response)

Reviewer #2: (No Response)

Reviewer #3: All comments have been addressed

Reviewer #4: All comments have been addressed

2. Is the manuscript technically sound, and do the data support the conclusions?

Reviewer #1: Yes

Reviewer #2: Partly

Reviewer #3: Yes

Reviewer #4: Yes

3. Has the statistical analysis been performed appropriately and rigorously? 

Reviewer #1: Yes

Reviewer #2: Yes

Reviewer #3: Yes

Reviewer #4: Yes

4. Have the authors made all data underlying the findings in their manuscript fully available?

Reviewer #1: Yes

Reviewer #2: Yes

Reviewer #3: Yes

Reviewer #4: Yes

5. Is the manuscript presented in an intelligible fashion and written in standard English?

Reviewer #1: No

Reviewer #2: Yes

Reviewer #3: Yes

Reviewer #4: Yes

6. Review Comments to the Author

Reviewer #1: 1. Rephrase to reflect a more inclusive and respectful tone e.g. Line 208 "unknon gender"  unreported

2. Line 212 "a month after the war"  If the exact timeline is important for the study, specifying the start date of the war or elaborating on how long the war had been going on by the time the study was conducted would provide greater clarity.

3. Table 1: The term 'No' under Higher Education is unclear. Does it refer to no formal education or a level lower than a bachelor's degree?

4. Table 1: The organizational variables should be fully detailed. For example, the term 'Information' remains unclear.

5. Line 227 "various social media networks" specifying which networks or types of networks (e.g., Facebook, WhatsApp) could offer better insight into how participants were targeted.

6. Line 313 occupational burnout in the war sample" could be simplified to "occupational burnout among participants" for smoother reading.

7. "The criterion variable" is used in one instance, and "the outcome variable" is used earlier. These terms should be consistent throughout.

8. Line 321  "results are presented in Tables 3" refer to "Table 3" (singular) for consistency

9. The terms "stress," "work-to-family conflict," "routine disruptions," and "sleep disruptions" are inconsistently capitalized.

10. The finding that "stress, work-to-family conflict, routine disruptions, and sleep disruptions" contributed significantly to burnout is important, but the magnitude of these effects (e.g., the beta coefficients or effect sizes) should be briefly mentioned.

11. The explanation of the non-significant findings regarding gender and family-to-work conflict is too brief. Offering a possible interpretation or context for why these variables did not significantly impact burnout could enhance understanding.

12. The term "work-to-family conflict" is abbreviated as "WFC" partway through the paragraph, but it is first presented in full. For consistency, either use the abbreviation throughout or introduce the abbreviation earlier.

13. The description of volunteer opportunities as moderating the relationship between WFC and burnout is vague. The phrase "as in higher volunteer opportunities the association...was insignificant" could be clearer. For example, more detail on how this moderation worked would enhance understanding. What exactly about the volunteer opportunities mitigated burnout?

14. The shift between past and present tense (e.g., "examined" vs. "are presented") feels inconsistent. Keeping the tense consistent throughout the passage would improve flow.

15. In the conclusion, outline specific directions for future research, such as how other types of organizational support could be explored or how different crisis settings (e.g., non-war-related crises) might yield different results.

Below are the key suggestions:

1. Clarify Key Terms and Concepts

2. Avoid redundancy in the discussion by consolidating similar points. For example, the relationship between routine disruption and burnout is mentioned multiple times. Streamlining this information would enhance readability.

3. Ensure tense consistency (e.g., switching between past and present tense in some sections) to maintain a professional and polished tone.

4. Simplify overly complex sentences to enhance clarity. Long, dense sentences can obscure the key message, so aim for concise language where possible.

5. Proofread for grammar and syntax errors, ensuring clear communication and a polished final draft.

Reviewer #2: Acceptable, however, please note that hypotheses in the literature review should be added because discussion and conclusion could be more precious and comprehensive.

Reviewer #3: (No Response)

Reviewer #4: The authors have addressed and responded my important comments as I raised; occupation and daily working hours, measuring results, and discussion section. I am appreciate for this revision.

7. PLOS authors have the option to publish the peer review history of their article (what does this mean?). If published, this will include your full peer review and any attached files.

Reviewer #1: No

Reviewer #2: No

Reviewer #3: No

Reviewer #4: No

---

## [Author Response · Author response to Decision Letter 1]

2 Nov 2024

Attached is a letter addressing the reviewers' comments.

---

## [Decision Letter · Decision Letter 2]

7 Nov 2024

PONE-D-24-34427R2Occupational burnout during war: The role of stress, disruptions in routine, sleep, work-family conflict, and organizational support as a moderatorPLOS ONE

Dear Dr. Kalfon Hakhmigari,

Thank you for submitting your manuscript to PLOS ONE. After careful consideration, we feel that it has merit but does not fully meet PLOS ONE’s publication criteria as it currently stands. Therefore, we invite you to submit a revised version of the manuscript that addresses the points raised during the review process.

We look forward to receiving your revised manuscript.

Kind regards,

Supaprawat Siripipatthanakul, Ph.D.

Academic Editor

PLOS ONE

Journal Requirements:

Additional Editor Comments:

(1) The abstract format for PloS One is as follows: background, methods, results, and conclusion (please add the subheadings).

(2) Flexibility in work hours, Consultation and support, Information about the organization, Security regarding stability, and Opportunities for volunteering could be other variables. If researchers would like to include these variables, Hypotheses could be changed (It is suggested to be deleted).

(3) Kindly check for sample size 373 or 374 (Table 1: total gender is 373).

(4) Lines 368 (Page 70, R2), To examined the first hypothesis, the mean level of occupational burnout among 315 participants was high (M=3.95, SD=1.15) (please check grammar). Descriptive analysis could not be used to develop hypotheses. The hypothesis is for inferential statistics.

(5) Table 2, please check for mean and SD. (five or seven-point rating scale, should not be over 5 or 7).

(6) Please check for hypotheses regarding Table 2 and Table 3. The hypotheses could be as follows;

H1: Stress has a positive effect on occupational burnout.

H2: Work-to-family conflict has a positive effect on occupational burnout.

H3: Family-to-work conflict has a positive effect on occupational burnout.

H4: Sleep disruption has a positive effect on occupational burnout.

H5: Routine disruption has a positive effect on occupational burnout.

(7) Pearson (Table 2) and Regression (Table 3) differ regarding gender. It is suggested that only one analysis, Pearson or Regression, be selected (or gender should be used as the moderator). Only Table 2 could better explain the hypothesis's results.

(8) Table 4 is ok but only includes some predictors; it is suggested to include H1-H5.

(9) Please recheck for Tables 3 and 4 regarding H1-H5.

(10) The conclusion in the abstract should be summarized for H1-H5, and

the last section of the conclusion should be supported or rejected for H1-H5. Thus, please revise to H1-H5 (gender could be the moderator).

Reviewers' comments:

Reviewer's Responses to Questions

**Comments to the Author**

1. If the authors have adequately addressed your comments raised in a previous round of review and you feel that this manuscript is now acceptable for publication, you may indicate that here to bypass the “Comments to the Author” section, enter your conflict of interest statement in the “Confidential to Editor” section, and submit your "Accept" recommendation.

Reviewer #1: (No Response)

2. Is the manuscript technically sound, and do the data support the conclusions?

Reviewer #1: Yes

3. Has the statistical analysis been performed appropriately and rigorously? 

Reviewer #1: Yes

4. Have the authors made all data underlying the findings in their manuscript fully available?

Reviewer #1: Yes

5. Is the manuscript presented in an intelligible fashion and written in standard English?

Reviewer #1: Yes

6. Review Comments to the Author

Reviewer #1: (No Response)

7. PLOS authors have the option to publish the peer review history of their article (what does this mean?). If published, this will include your full peer review and any attached files.

Reviewer #1: No

---

## [Editor Report · Decision Letter 3]

19 Dec 2024

Occupational burnout during war: The role of stress, disruptions in routine, sleep, work-family conflict, and organizational support as a moderator

PONE-D-24-34427R3

Dear Dr. Kalfon Hakhmigari,

We’re pleased to inform you that your manuscript has been judged scientifically suitable for publication and will be formally accepted for publication once it meets all outstanding technical requirements.

Kind regards,

Supaprawat Siripipatthanakul, Ph.D.

Academic Editor

PLOS ONE

Additional Editor Comments (optional):

The revised version is acceptable.
---

## [Editor Report · Acceptance letter]

26 Dec 2024

PONE-D-24-34427R3 

PLOS ONE

Dear Dr. Kalfon Hakhmigari, 

I'm pleased to inform you that your manuscript has been deemed suitable for publication in PLOS ONE. Congratulations! Your manuscript is now being handed over to our production team.

Kind regards, 

on behalf of

Professor Supaprawat Siripipatthanakul 

Academic Editor

PLOS ONE